# Improving the Aromatic Profiles of Catarratto Wines: Impact of *Metschnikowia pulcherrima* and Glutathione-Rich Inactivated Yeasts

**DOI:** 10.3390/antiox12020439

**Published:** 2023-02-10

**Authors:** Vincenzo Naselli, Rosario Prestianni, Natale Badalamenti, Michele Matraxia, Antonella Maggio, Antonio Alfonzo, Raimondo Gaglio, Paola Vagnoli, Luca Settanni, Maurizio Bruno, Giancarlo Moschetti, Nicola Francesca

**Affiliations:** 1Department of Agricultural, Food and Forest Sciences (SAAF), University of Palermo, Viale delle Scienze Bldg. 5, Ent. C, 90128 Palermo, Italy; 2Department of Biological, Chemical and Pharmaceutical Sciences and Technologies (STEBICEF), University of Palermo, Viale delle Scienze, Parco d’Orleans II, Bldg. 17, 90123 Palermo, Italy; 3Lallemand Italia, Via Rossini 14/B, Castel D’Azzano, 37060 Verona, Italy

**Keywords:** alcoholic fermentation, Catarratto grape variety, glutathione, *Metschnikowia pulcherrima*, VOC’s, wine aroma

## Abstract

Catarratto is one of the most widely cultivated grape varieties in Sicily. It is an indigenous non-aromatic white grape variety. Despite its widespread use in winemaking, knowledge of the aroma and chemical and microbiological properties of Catarratto wines is quite limited. The influence of *Metschnikowia pulcherrima* combined with *Saccharomyces cerevisiae* on the aromatic expression of Catarratto wines was investigated with and without the addition of glutathione-rich inactivated yeast. The substance is a natural specific inactivated yeast with a guaranteed glutathione level used to limit oxidative processes. The aromatic profiles of the final wines were determined through analysis of the volatile organic compounds using a solid-phase microextraction technique that identified 26 aromatic compounds. The addition of *M. pulcherrima* in combination with the natural antioxidant undoubtedly increased the aromatic complexity of the wines. Dodecanal was exclusively detected in the wines processed with glutathione-rich inactivated yeasts. Furthermore, the presence of this natural antioxidant increased the concentration of six esters above the perception threshold. Sensory analysis was also performed with a panel of trained judges who confirmed the aromatic differences among the wines. These results suggest the suitability of glutathione-rich inactivated yeasts for determining the oxidative stability of Catarratto wines, thus preserving its aromatic compounds and colour.

## 1. Introduction

Identifying new strategies for the aromatic enhancement of wine produced from non-aromatic grape varieties is one of the major objectives of oenological microbiology research. During alcoholic fermentation (AF), yeast metabolic activity and winemaking techniques determine the biosynthesis of several products that influence wine aroma [1,2]. The application of non-conventional yeasts isolated during the fermentation of traditional fermented beverages represents an alternative for producing a variety of alcoholic beverages [3], including wine.

Raw materials with a high sugar content if subjected to spontaneous fermentation can provide potential starters with interesting traits. Wild *Saccharomyces* and non-*Saccharomyces* spp. may generate flavour profiles with desirable characteristics to be applied at industrial level [4]. Several authors have isolated strains of *Saccharomyces cerevisiae* from high-sugar-containing matrices such as manna, honey and honey by-products [5,6,7,8] that were successfully applied in experimental Catarratto cultivar winemaking [1].

Recently, the controlled inoculation of selected non-*Saccharomyces* and *S. cerevisiae* strains has permitted the production of higher quality wines. The current trend is to exploit non-*Saccharomyces*/*Saccharomyces* sequential inoculation to achieve a positive impact in terms of aroma [9]. The result is closely related to the species of the multi-starter cultures involved in the sequential inoculum [9].

The final aroma of wines can be modulated not only by *Saccharomyces* but also by non-conventional yeasts. The metabolic impact of non-*Saccharomyces* yeasts during the early stages of fermentation is sufficient to trigger significant changes to the wine’s volatile profile; they are suitable for inoculation as co-starters with strains of *S. cerevisiae* [10].

Fermentation processes using mixed strains with the sequential addition of non-*Saccharomyces* and *S. cerevisiae* strains tend to reproduce what happens naturally during spontaneous wine fermentation concerning population dynamics [11]. Indeed, the levels of non-*Saccharomyces* yeast populations are reduced over time, leaving space for *S. cerevisiae* to dominate and conclude the AF [12].

The use of non-*Saccharomyces* yeasts has improved the primary aromas of wines, as the production of specific enzymes enables the precursors present in the must to release volatile molecules. Their activity also affects secondary aromas through the production of volatile organic compounds (mainly alcohols and esters) that can influence typical aromatic expressions such as fruity notes [11].

The impact determined by sequential inoculation can influence various aspects of wine characteristics. In recent oenological studies, strains belonging to the species *Lahancea thermotolerans* and *Starmerella bacillaris* in sequential inoculation with *S. cerevisiae* achieved wines with a significant amount of lactic acid and glycerol, respectively, while strains of *Torulospora delbrueckii* and *Hanseniaspora uvarum* in mixed cultures and in sequential inoculation with *S. cerevisiae*, on the other hand, seemed to influence the composition in terms of higher alcohol and ester contents [9].

Among the non-*Saccharomyces* yeasts, *Metschnikowia pulcherrima* is one of the species most abundant in the initial phase of AF of grape musts. In mixed cultures with *S. cerevisiae*, *M. pulcherrima* rapidly declines due to its low resistance to the ethanol produced by *S. cerevisiae* [13]. Some strains of *M. pulcherrima* are known to synthesize fruity esters and can increase the concentrations of terpenes or thiols generally masked by higher alcohols [14]. Non-aromatic grape varieties lack varietal aromatic precursors (terpenes or thiols) and the presence of fruity aromas (pineapple) due to the fact that an increased ethyl octanoate content determines a positive sensorial impact. Some thiol precursors such as 4-methyl-4-sulfanylpentan-2-one, as well as those produced by *S. cerevisiae*, can be synthesized by *M. pulcherrima* at much higher concentrations, thus significantly influencing the characteristics of wine [15]. Recently, *M. pulcherrima* was successfully used in a sequential inoculation with *S. cerevisiae* for a reduction in ethanol content in Merlot wines [16] and to improve the aromatic complexity in Shiraz and Cabernet Sauvignon wines [17].

A solution aimed at limiting the loss of aromaticity in white wines is represented by the addition of a natural antioxidant such as glutathione at the beginning of the vinification process [18]. The application of glutathione during winemaking has positive effects on the colour and aroma stability of white wines [19]. The glutathione content naturally present in musts is relatively low, and its quantities are closely related to the reactions that characterise the fermentation process as well as the metabolic activities of the yeasts [16]. Its use in oenology provides considerable advantages as its antioxidant activity is capable of limiting browning in white grape must as it inhibits polyphenol polymerisation and severely limits the production of compounds such as sotolone that give wine a fenugreek or curry odour [20]. Glutathione’s degree of protection also extends to the aromatic molecules in wines, especially the esters, volatile thiols and terpenes produced by yeasts during alcoholic fermentation that are present in greater quantities when glutathione is added to the must [21]. Some sulphite-free wines are produced by exploiting the antioxidant activity of glutathione in place of potassium metabisulphite, meeting the needs of consumers who are more sensitive to the negative health effects of sulphur dioxide [22].

This study focused on the potential of non-*Saccharomyces* and *Saccharomyces* yeasts for the aromatic improvement of wine produced from non-aromatic grape varieties such as Catarratto. We evaluated the sequential inoculation of a commercial non-*Saccharomyces* yeast strain (*M. pulcherrima*) and *S. cerevisiae* SPF52 isolate from honey by-products to simulate what would occur during spontaneous fermentation. Secondly, we assessed the ability of exogenous glutathione addition during fermentation in the form of inactivated yeast to influence the technological and aromatic properties of wine.

The aims of this research were to investigate: (i) the impacts of *M. pulcherrima* associated with *S. cerevisiae*; (ii) the effect of an antioxidant compound on the aroma and sensory profiles of Catarratto wine; and (iii) the volatile organic compound composition of Catarratto white wine.

## 2. Materials and Methods

### 2.1. Experimental Drawing and Sampling

The experimental design (Figure 1) was composed of four treatments: T1, sequential inoculum of FLAVIA^®^ MP346/*S. cerevisiae* SPF52; T2, the use of Glutastar^TM^ to the bulk must and sequential inoculum with FLAVIA^®^ MP346/*S. cerevisiae* SPF52; C1, single inoculum of *S. cerevisiae* SPF52; C2, the addition of Glutastar^TM^ and fermentation by *S. cerevisiae* SPF52.

FLAVIA^®^ MP346 is a pure culture of *M. pulcherrima* selected by the Universidad de Santiago de Chile (USACH) for its specific capacity to release enzymes with arabinofuranosidase activity [23]. Glutathione-rich inactivated yeast (GIY) is an inactivated yeast mass with a guaranteed glutathione level [24]. GIY and FLAVIA^®^ MP346 were provided by Lallemand Inc. (Castel D’Azzano, Verona, Italy). The *S. cerevisiae* SPF52 strain used in this study belonged to the yeasts collection of the Department of Agricultural, Food and Forestry Sciences (SAAF; University of Palermo, Italy); it was isolated from fermented honey by-products [6] and selected for its high performance in fermenting Catarratto grape must [1].

Samples were collected from clarified bulk must just after the inoculum of *M. pulcherrima* MP346, after the inoculation of *S. cerevisiae* SPF52, during AF (day 3, 6, 12 and 18), during ageing in a steel tank (1, 3 and 5 months) and at bottling. All samples were transported at 4 °C into a portable fridge and subjected to analysis within 24 h after collection.

### 2.2. Winemaking

After hand harvesting, grapes were stemmer-crushed and treated with 2 g/hL of potassium metabisulphite (Chimica Noto s.r.l., Partinico, Italy). Clarification of the must was carried out at 4 °C for one day by using pectolytic enzymes [Lallzyme^®^ C-Max (Lallemand Inc. Italia, Castel D’Azzano, Verona, Italy); dosage: 4 g/hL].

T1 and T2 were inoculated with FLAVIA^®^ MP346 at 25 g/hL when the clarified must had reached a temperature of 16 °C. The strain *S. cerevisiae* SPF52 was used in a liquid concentrated form [about 7.00 × 10^10^ colony-forming units (CFU)/g].

After 24 h, T1 and T2 were inoculated with *S. cerevisiae* SPF52 (20 g/hL), while the controls, C1 and C2, were inoculated immediately with the SPF52 strain at the same dose. Before the inoculum of the starter yeast, GIY (40 g/hL) was added to treatments T2 and C2. The organic nutrient Stimula Chardonnay^TM^ (SC; Lallemand Inc. Italia, Castel D’Azzano, Verona, Italy) was added to all tanks (40 g/hL) prior to *S. cerevisiae* yeast inoculation. The use of Stimula Chardonnay^TM^ with *S. cerevisiae* SPF52 was chosen because of the results obtained by previous vinifications on Catarratto wines [1]. The fermentation was carried out at 18 °C in 12 steel tanks with a volume of 2.5 hL each. At the end of AF, the wines were cold-settled, their yeast lees were racked off and they were transferred into stainless-steel tanks at 15° C and topped with nitrogen to avoid oxidation until bottling. During ageing, malolactic fermentation was prevented by keeping the free SO_2_ values above 35 mg/L until bottling. Tartaric stability was ensured through the addition of 8 g/hL of metatartaric acid (Chimica Noto s.r.l., Partinico, Italy). Each treatment was performed in triplicate.

### 2.3. Monitoring Yeast Populations

During the AF, all must samples were microbiologically analysed to determine the total yeast concentration (TY) using the protocol described by Pallmann et al. [25]. *Saccharomyces* and non-*Saccharomyces* yeasts colonies were distinguished as reported by Valera [3]. The analyses were conducted in triplicate.

### 2.4. Yeast Collection and Genotypic Characterization

Yeasts were isolated from WL medium, purified on the same medium and then subjected to morphological analysis, as reported by Pallmann et al. [25], and genotypic characterisation.

Genomic DNA for PCR assays was prepared from yeast isolates after growth in YPD broth media at 25 °C for 48 h. Cells were harvested, and DNA was extracted using the InstaGene Matrix kit (Bio-Rad Laboratories, Hercules, CA, USA) according to the manufacturer’s instructions. According to Sinacori et al. [8], yeasts were discriminated by RFLP of the region spanning the internal transcribed spacers (ITS1 and ITS2) and the 5.8S rRNA gene. Species-level identification of each group was confirmed by sequencing the D1/D2 region of the 26S rRNA gene following the procedure described by Guarcello et al. [7]. DNA sequencing reactions were performed at AGRIVET (University of Palermo, Italy). Sequences were manually corrected using Chromas 2.6.2. (Technelysium Pty Ltd., Brisbane, Australia). Nucleotide sequences were compared to GenBank sequences through BLASTn searches.

### 2.5. Dominance of S. cerevisiae and M. pulcherrima Isolates

The dominance of the inoculated *S. cerevisiae* and *M. pulcherrima* was verified as reported by Legras et al. [26] and Barbosa et al. [27]. Fingerprinting profiles were analysed as reported by Alfonzo et al. [28].

### 2.6. Must and Wine Analysis

#### 2.6.1. Chemical Properties

Chemical properties such as sugars (glucose and fructose, g/L) and residual sugars (g/L), yeast-assimilable nitrogen (ammoniacal nitrogen and alpha-amino nitrogen, g/L), organic acids (malic acid, lactic acid and acetic acid, g/L), glycerol (g/L) and ethanol (% *v*/*v*) were quantified during and at the end of the AF using the methods described by Prestianni et al. [29].

The pH values were measured using a pH 70 Vio FOOD pH meter (XS Instruments, Carpi, Italy), and total acidity (g/L of tartaric acid) was detected through the procedure proposed by OIV-MA-AS313-01 [30]. Free and total sulphur dioxide was determined in accordance with Alfonzo et al. [1].

The analysis of the chemical composition of wines analysed included ash alkalinity, buffering power, total extract, total phenols, flavans reactive to 4-(dimethylamino)cinnamaldehyd, oxidation tests, total phenols and extracts were performed as reported by Alfonzo et al. [1].

#### 2.6.2. Volatile Organic Compounds

All reagents were of analytical grade. Ethyl benzoate was purchased from Sigma-Aldrich (82024 Taufkirchen, Germany). n-Alkane standards (C8 to C40) were purchased from Aldrich Chemical Co. (St. Louis, MO, USA).

An automatic SPME holder (Supelco^®^, Bellefonte, PA, USA) was used for evaluation of VOC profiles. A fiber 50/30 µm divinylbenzene (DVB)/carbowax (CAR)/polydimethylsiloxane (PDMS) of 1 cm length was used for fractionation of volatile compounds from the headspace (HS) of the conditioned wines. Prior to its use, the fiber was conditioned for 1.5 h at 250 °C in the inlet of the gas chromatograph according to Supelco^®^ Co. Analysis of wine aroma was performed following a slightly modified method proposed by Sagratini et al. [31]. For extraction, each aliquot (10 mL) of the wine samples and 2.2 g of NaCl were placed into a 20 mL vial (75.5 × 22.5 mm) (Supelco, Bellefonte, PA, USA). The samples were equilibrated at 35 °C for 15 min, stirring at 600 rpm. The SPME fiber was exposed to the wine samples for 30 min in the headspace of the sample kept at 35 °C. The flavour compounds were desorbed for 5 min from the fiber to the column through a splitless injector at 250 °C. The SPME fibres were cleaned to prevent cross-contamination by inserting the fibre into the auxiliary injection port at 250 °C for 30 min and were then re-used. All samples were prepared and analysed in triplicates in standard 20 mL volume headspace vials.

Semi-quantification of volatile compounds was performed using an Agilent 7000C GC system fitted with a fused silica apolar DB-5MS capillary column (30 m × 0.25 mm i.d.; 0.25 μm film thickness) (Santa Clara, CA, USA) coupled to an Agilent triple quadrupole Mass Selective Detector MSD 5973. The ionization voltage was 70 eV, the electron multiplier energy was 2000 V and the transfer line temperature was 270 °C. The solvent delay was 0 min. Helium was the carrier gas (1 mL/min). The temperature programme was from 35 °C (0 min) to 270 °C at 8 °C min^−^^1^, from 270 °C (2 min) to 300 °C at 15 °C min^−^^1^ and then 300 °C for 5 min. Volatile compounds were injected at 250 °C automatically in the splitless mode. Linear retention indices were calculated using *n*-alkanes as reference compounds. For the analysis of alkane solutions (C_8_-C_40_) (Sigma-Aldrich, USA), the injector mode was set in the 10:1 split mode. The individual peaks were analysed using the GC-MSolution package, version 2.72. Identification of compounds was carried out using the Adams, NIST 08, Wiley 9 and FFNSC 2 mass spectral databases.

For each volatile organic compound identified, the odour activity value (OAV) as described by Butkhup et al. [32] was calculated in order to assess which VOCs contributed significantly to the odour series characterising each wine.

### 2.7. Sensory Analysis

A total of 15 judges (7 women and 8 men, ranging from 25 to 46 years old) with previous experience in wine tasting participated in the evaluation of the sensory profile of the wines carried out as described by Jackson [33]. The judges were subjected to preliminary tests to determine their sensory performances in terms of their basic taste and the aromas associated with the wines. The sensory profiles of the wines obtained from Catarratto grapes were constructed using two selected panels each of ten judges trained over several sessions. The fifteen panellists compared the four experimental wines during different sessions. They consensually generated 36 sensory descriptive attributes for appearance, odour, flavour, taste, overall quality and finish in several sessions. The set of attributes were: appearance (green reflexes and yellow colour); odour (banana, citrus, fatty, floral, fruity, grape, green almond, intensity, pear, persistence, pineapple and sweet fruit); taste (bitter, salty, sour and sweet); mouthfeel (body or balance); flavour (banana-like, cherry pit, citrus, fruity, intensity, mandarin orange, persistence, pineapple, sweet apple and sweet fruit), overall quality (flavour, mouth-feel, odour and taste) and finish (after-smell and after-taste). The different descriptors were quantified using a 9-point intensity scale as reported by Alfonzo et al. [28].

The sensory test was carried out following the procedures described by Alfonzo et al. [1].

### 2.8. Statistical Analysis

In order to determine statistically significant differences between the properties monitoring during the AF (chemical and technological data) and in the final wines (sensory analysis and VOCs composition), the ANOVA test was applied. Tukey’s test was used for multiple mean comparisons (statistical significance: *p* < 0.05).

Multiple factor analysis (MFA) was carried out in order to distinguish the different treatments from the data acquired during the sensory analysis following the methodology reported by Alfonzo et al. [1]. Agglomerative hierarchical clustering (AHC) was performed to group the trials according to their dissimilarity, as measured by Euclidean distances and Ward’s method.

In order to assess the existing correlation between the aromas detected during the sensory analysis and the VOCs with an odour activity value > 1, a principal component analysis (PCA) was performed using the XLstat software version 2019.2.2 (Addinsoft, New York, NY, USA) for Excel.

## 3. Results and Discussion

### 3.1. Microbial Growth Dynamic

The concentrations of yeasts (presumptive *Saccharomyces* (PS), non-*Saccharomyces* (NS) and presumptive *Metschnikowia* (PM)) during the alcoholic fermentation (AF) are shown in Figure 2. The PS and NS levels in the Catarratto must were around four logarithmic cycles (Figure 2a,b), while no isolates attributable to the genus *Metschnikowia* were detected (Figure 2c). Catarratto musts are usually poor for the presence of indigenous *Metschnikowia* spp., although in musts from Sicilian Catarratto grapes, *M. pulcherrima* has been isolated at percentages ranging from 0.2 to 1.1% [34].

The *M. pulcherrima* MP346 inoculum concentration in T1 and T2 was close to 6.5 Log CFU/mL. The concentration of PS after the SPF52 inoculum ranged from 7.3 (T1) to 7.6 (C1) Log CFU/mL in all treatments. On day 3 of AF, PS showed an increase to 7.4–8.0 Log CFU/mL for all trials. The NS populations were lower and in the range of 2.3–3.2 Log CFU/mL. The reduction in the NS yeast populations during AF is a known phenomenon attributable to several causes such as metabolite production by *S. cerevisiae*, nutrient limitation and low resistance to ethanol [13]. The PM levels were 3.0 Log CFU/mL for T1 and 4.6 Log CFU/mL for T2 after 3 days of AF and were lower than the limit of detection in the C1 and C2 samples. Indeed, the lower microbial load of the PM populations observed at 3 days of AF in T1 and T2 compared with C1 and C2 could be due to the lower ethanol concentration detected in T1 and T2 (Appendix A). At day 6 of AF, when the ethanol reached concentrations above 6% *v*/*v*, the PS values reached levels in the range of 7.0–8.0 Log CFU/mL, whereas both NS and PM were undetectable in any trials. The absence of *M. pulcherrima* in trials inoculated with the commercial preparation FLAVIA^®^ MP346 (T1 and T2) could be due to the above-mentioned factors. Some authors have recorded a significant decrease in the concentration *of M. pulcherrima* after 9 days of AF when sequential inoculation with *S. cerevisiae* occurred [15].

From the 12th day until the end of AF (18 d), the PS populations decreased slightly from 7.3–8.0 to 6.7–7.0 Log CFU/mL in all treatments. The microbiological count values for *S. cerevisiae* were found by Scacco et al. [35] on Sicilian Catarratto musts inoculated with selected starter strains of the same species.

### 3.2. Molecular Analysis

In relation to the macro- and microscopic characteristics, 949 colonies were analysed; from these, 592 isolates showed the typical characteristics of yeasts belonging to the *Saccharomyces* genus. The amplicon size of the 5.8S-ITS region was around 850 bp and confirmed the presumptive species identity of *S. cerevisiae* for all isolates. The other isolates (n = 357) were assigned to the NS yeast group.

A total of 233 isolates were morphologically identified as *Metschnikowia* spp. and showed an ITS amplicon between 380 and 400 bp. The ITS amplicon sizes were equivalent to those reported in the literature for *M. pulcherrima* [36]. The isolates of the PS group (n = 592) and PM (n = 233) were characterised by RFLP analysis of the 5.8S-ITS region.

The PS RFLP profiles were similar to those indicated by Granchi et al. [37]. Consequently, the PS group represented putative *S. cerevisiae*. The sizes of the RFLP profiles of the PM were equivalent to those described in the literature for the species *M. pulcherrima* [37].

The different profiles may have been caused by the presence of native *S. cerevisiae*, although less representative, being present among the isolates obtained. Indeed, the PS count values detected before SPF52 inoculation (4.1 Log CF/mL) clearly explain the presence of eight additional interdelta profiles. The interdelta profile of *S. cerevisiae* SPF52 was the most frequently (>93%) isolated. The strain typing of *M. pulcherrima* was carried out by RAPD-PCR. The results from these analyses showed that all the 233 isolates represented a unique strain.

The genotypic identification of the yeasts was completed by pairwise alignment of the D1/D2 sequence with the type of strain of each species (*S. cerevisiae* CBS 1171^T^ and *M. pulcherrima* CBS 5833^T^). A comparison of the sequences of the D1/D2 region of the two reference strains showed a 100% similarity to the sequences of the type strains of each species, confirming the identification obtained by the RFLP analysis.

### 3.3. Kinetics of the Main Oenological Properties

The fermentations carried out in the presence of *S. cerevisiae* SPF52 as the only inoculated strain (C1 and C2) and the corresponding trials with *M. pulcherrima* (T1 and T2) were able to conclude the AF as determined by the complete consumption of sugars.

The trends of the principal oenological data during AF are shown in Appendix A. The fermentation was concluded in 18 days on average.

After 3 d of AF, differences in pH, TA and the concentrations of sugars, ethanol, ammonia nitrogen and alpha amine nitrogen were observed among the trials. The highest differences in the sugar, glycerol and ethanol contents were registered at day 6 of AF. Specifically, C2 showed the lowest values in residual sugars (58.79 g/L); glucose was 25.80 g/L and fructose was 32.99 g/L, and consequently, it showed the highest values of ethanol (8.44\% *v*/*v*). The glycerol contents observed in T1, T2 and C2 were similar (5.19–5.28 g/L), whereas the lowest values were found in C1 (5.06 g/L). This trend was observed until the 12th day of AF.

At the end of AF, the glucose concentrations ranged from 1.10 (T1 and T2) to 1.62 g/L (C1), whereas the fructose concentration was slightly higher and within the range of 1.39–2.60 g/L. No differences were observed for TA, whereas VA’s values ranged from 0.27 (T1) to 0.31 (C1 and C2) g/L acetic acid. The pH values varied between treatments, where T1 and T2 had slightly lower values (3.41 and 3.43, respectively) when compared to both the control trials C1 (3.47) and C2 (3.51). The ethanol concentrations ranged between 11.35 and 11.43% (*v*/*v*); the comparison between the T1 and T2 and the C1 and C2 treatments showed no significant differences. In contrast, Contreras et al. [38] reported that some strains of *M. pulcherrima* are able to decrease the amount of ethanol by as much as 1% (*v*/*v*) during fermentation. An analysis of the ethanol production during AF a revealed lower ethanol production in the T1 and T2 trials after 3 d of AF. After AF, differences in the ethanol concentration between the different trials were not statistically significant. This phenomenon could be attributable to the presence of *M. pulcherrima* up to the 3rd day of AF (3.0–4.6 Log CFU/mL).

The malic acid levels decreased in all the treatments from an initial concentration of 1.90 g/L in the must to 1.28–1.50 g/L at the end of AF. Contrary to the reports of Ruiz et al. [15], no decreases were recorded in T1 and T2 compared to C1 and C2, although these authors showed that in fermentations conducted with *M. pulcherrima*/*S. cerevisiae*, a decrease in the malic acid content of 0.2 g/L occurred in the wines. Lactic acid was absent in all the trials. The highest concentration of glycerol was found in C2 (6.57 g/L), and lower values (>5 g/L) were detected in the other wines. In this case, the sequential inoculum with *M. pulcherrima*/*S. cerevisiae* did not produce an increase in the glycerol concentration, in contrast to what has been observed in white wines made with the Verdejo variety [15].

During the five months of ageing in stainless steel tanks, there were no substantial changes in the monitored chemical properties (Appendix A). There was a decrease in residual sugars, glucose and fructose, and all the other properties remained constant or showed minimal variations.

### 3.4. Oenological Data Analysis

The values of the physico-chemical properties of the wines are reported in Table 1.

The free and total SO_2_ values were variable in the different wines. In particular, the highest free SO_2_ values were observed in T1 and C2 (29 mg/L), while the highest total SO_2_ value was observed in T1 (128 g/L).

The total extract was higher than the minimum legal values, which for white wines are fixed at >14 g/L [39]. In this study, all the wines exceeded this threshold; the values were in the range of 18.50–19.10 g/L for C1 and T2, respectively, which were comparable to the results described in Scacco et al. [35] on Sicilian Catarratto wines.

The T1 and T2 trials retained a greater susceptibility to undergo oxidation than the C1 and C2 controls, which was independent of the use or non-use of GIY with oxidation test values of 5.74 and 1.12% (T1 and T2) and 0% (C1 and C2). The presence of *M. pulcherrima* therefore appeared to exert a bio-protective action by predicting oxidations at the pre-inoculation of *S. cerevisiae*. The decrease in polyphenols was not due to the synthesis of polysaccharides by *M. pulcherrima* but to its bioprotective and inhibiting action against grape tyrosinases. In fact, in the pre-fermentative stage in the C1 and C2 controls, the absence of *M. pulcherrima* favored a significant increase in the optical density at 420 nm. At the same time, in the same controls there would have been a significant decrease in the total polyphenols resulting from the decrease in the phenolic class of the ortho-diphenols detected by means of the p-DACA reagent. The total polyphenol content was independent of the presence/absence of GIY. The null POM test values observed in the controls C1 and C2 may be due to a series of oxidation reactions of polyphenolic compounds that not even the addition of GIY in T2 was able to limit. The colonisation of the must by *M. pulcherrima* in the pre-fermentation phase probably led to a reduction in oxidative activities [40].

Regarding buffering power, there were negligible variations, and only the wine C2 reached statistically significant values compared to the other trials. The highest buffering power value was in C2 (32.34 meq/L), which was comparable to those reported in the literature in Sicilian Catarratto wines [40]. This was similar for ash alkalinity, where C2 had the highest value (13.43 meq/L); the wine values were within the range of 11–17 meq/L, which were similar to those reported in the literature for white wines [41].

### 3.5. Volatile Organic Compound Composition

The samples showed differences mainly at the quantitative level. Twenty-six compounds were detected, and they were grouped into several classes: alcohols, ethers, aldehydes, ethyl esters of fatty acids (EEFAs), higher alcohol acetates (HAAs), ethyl esters of branched acids (EEBAs), miscellaneous esters (MEs) and other compounds. For clarity, the classification of esters was reported as described by Alfonzo et al. [1]. The most-concentrated compounds in all the samples were EEFAs (2318.98–1401.74 ppb) followed by MEs (233.83–98.84 ppb) and alcohols (36.48–18.84 ppb).

The must inoculated with *M. pulcherrima* MP346 produced less alcohols than the controls. 3-methyl-1-butanol and phenylethyl alcohol were the compounds detected in the highest quantity in C2. A similar condition was observed in Riesling wines fermented by sequential inoculation with *M. pulcherrima*/*S. cerevisiae* [14].

The compound most commonly detected in the aldehyde class was dodecanal. In the wines produced in the absence of GYI, it reached a maximum concentration in C1 (11.06 ppb). Aldehydes, particularly decanal and dodecanal if they are present in high concentrations, can result in the appearance of an unpleasant “green” odour in wines [42].

Esters directly and indirectly influence wine aroma by means of highly varied interactions. The fermentation process applied significantly influences the quality and quantity of esters [43,44]. The wine samples inoculated with *M. pulcherrima* MP346 showed a higher content of esters (2318.98 ppb in T1 and 2056.15 ppb in T2) than the controls (1401.74 ppb in C1 and 1848.45 ppb in C2). Among the esters, the most representative was ethyl decanoate, which was produced in amounts over 1000 ppb in the wines inoculated with *M. pulcherrima* MP346. Indeed, in Riesling musts inoculated with the same commercial strain of *M. pulcherrima*, the quantities detected were half of those present in the Catarratto musts [14,45].

The ethyl decanoate content reported by Benito et al. [14] and Mislata et al. [45] does not appear to have been impacted by the presence of *M. pulcherrima* MP346. However, in the Catarratto wines in this study, the levels of ethyl decanoate were significantly higher in the fermented wines with sequential inoculum.

A different situation was observed for ethyl octanoate, where the second EEFA was detected in greater quantities. Higher levels of ethyl octanoate were found in the experimental wines C1 and T1 without the addition of GIY. The effect of the glutathione-enriched inactivated yeast on ethyl octanoate was unclear, although these highly volatile hydrophobic esters exhibit significant variations in wines containing yeast-derivative products [46]. Among the 2-phenylethyl esters, two opposite situations were found for 2-phenylethyl hexanoate, which was detected only in C1 and C2, while 2-phenylethyl acetate was present exclusively in T1 and T2.

The determination of VOCs in the different wines is reported in Table 2.

However, the 2-phenylethyl acetate concentrations were lower than those determined for Riesling wines produced using *M. pulcherrima* MP346. Most likely, the strain of *S. cerevisiae* used as the starter for AF significantly influenced the levels of this ester [59].

Among the twenty-six VOCs, only seven compounds showed an OAV greater than 1 (Table 2), i.e., one aldehyde (dodecanal) and six esters (ethyl exanoate, ethyl octanoate, ethyl decanoate, ethyl 9-decenoate, 3-methyl-1-butanol acetate and methyl benzoate). Esters represent a group of compounds of considerable importance that are formed during AF through yeast metabolism and have a strong impact on the aromatic profile of wine [60].

### 3.6. Sensory Analysis

The data from the sensory evaluation are shown in Table 3. The trials revealed some differences correlated with the presence/absence of *M. pulcherrima* MP346 and GIY. 

The wines showed variability in terms of the attributes that defined appearance. The yellow colour values were in the range of 7.15–7.29, whereas, the green reflexes ranged between 3.63–4.04. The yellow colour values observed were higher than those shown by Scacco et al. [35], while the ratings associated with the green reflections attribute were similar. 

The T2 sample displayed a high score for 13 descriptors. The *M. pulcherrima* MP346 and GIY wine (T2) had the highest overall quality score (8.80). With regards to the odour attributes, the T1 and T2 wines showed the highest values for intensity and persistence, respectively. In addition, the T1 wine showed high scores for grape, fruity and fatty odours, the C1 wine showed high scores for citrus, floral, green almond and pineapple odours and the C2 wine was characterised by the presence of odours associated with banana, pear, pineapple and sweet fruit. The T2 wine was characterised by odour attributes with intermediate scores. In wines to which GIY was added (C2 and T2), citrus and floral odours were not perceived. Nevertheless, banana, citrus, floral, fruity and pear aromas were present in the Catarratto wines reported by Scacco et al. [35] but at lower levels. 

The descriptors associated with taste enabled discrimination of the wines. T1 and T2 showed high scores for sour flavours, whereas salty flavours showed high values in T2. In terms of mouthfeel, the T2 wine achieved high values for the body and balance attributes. No unpleasant odours or flavours were revealed for all the wines. The GIY increased the flavour intensity and persistence, confirming the results described by Alfonzo et al. [1]. Indeed, the treatment with GIY in combination with *M. pulcherrima* MP346 significantly improved the aromatic complexity of the T2 wine.

The T2 wine showed high intensity and persistence scores for flavours. The sensory descriptors with high flavour values were pineapple (C1), sweet fruit (C2) and fruity (T1 and T2). The T2 wine also excelled compared to the other wines for after-smell (8.50) and after-taste (8.71).

Correlations of the sensory analyses were examined by MFA. The number of sensory attributes (thirty-six variables) for the four wines made it possible to define two factors with an Eigen > 1 that represented a total variance of 89.64%. The correlation between the variables and the MFA factor was expressed by the value of the contribution and cos^2^. The incidence of the factors F1 (56.41%) and F2 (33.23%) on the total variance discriminated the different wines. Examining the loading plot (Figure 3), eight variables were located in both quadrants I and IV, ten were located in quadrant II and eleven were located in quadrant III.

Figure 4 reveals that the wines were clustered into three groups. In Figure 4a (MFA) and Figure 4b (AHCA), it is possible to observe how T1 and T2 represented a unique cluster. Interestingly, trial C1 did not cluster with trial C2. Indeed, the C1 and C2 trials represented a different cluster.

### 3.7. Sensory Profiles Associated with Volatile Organic Compounds

A PCA was used to evaluate the correlation between VOCs and aroma attributes. According to Figure 5, the F1 factor contributed 66.11% of the total variance, whereas the F2 factor explained 28.60% of the total variance.

Each wine, as can be seen from the biplot graph, was separate from the others. The C1 wine was associated with methyl benzoate, which produced green almond aromas [61]. A sensory analysis confirmed this attribute, and the highest scores were achieved in this trial. Ethyl 9-decenoate was the compound closely correlated with the T1 wine. This ester produces fruity and fatty odours [62], which were also detected in the sensory analysis, with the scores of fruity being higher than fatty. The grape aroma emitted by ethyl decanoate [63] represented the T2 wine. The highest sensory analysis attributes detected in the T2 wine were fruity and sweet fruit, and the grape aroma showed modest values. However, fruity and grape aromas are also associated with the presence of ethyl decanoate [64]. Finally, the C2 wine was closely associated with four odour descriptors (pineapple, sweet fruit, banana and pear). Only 3-methyl-1-butanol acetate and ethyl hexanoate were above the odour threshold and were responsible for the odours detected in the C2 wine by sensory analysis [61]. 

The imperfect correlation between the highest OAV values of VOCs and the sensory analysis might be attributable to the synergistic interaction of odour molecules (high OAVs with low OAVs) with each other. As a result, the odours related to specific compounds were absent or very slightly perceived during the sensory analysis.

## 4. Conclusions

In this research, four treatments were examined in order to investigate the effect of *M. pulcherrima* and an antioxidant on the aroma and sensory profile of Catarratto wines. The use of *S. cerevisiae* SPF52 from a non-winemaking origin confirmed that yeasts from honey and its derivatives can potentially be used as starter strains in oenology. The combined use of *M. pulcherrima* MP346 and GIY had a positive impact on the taste–olfactory complexity of the wines. These differences were also confirmed by a sensory analysis. The VOC profiles generated by the wines obtained in the presence/absence of *M. pulcherrima* MP346 were correlated to the addition of GIY from the point of view of the quantity–intensity effect.

Dodecanal was only detected in the wines without GIY, whereas six esters had an OAV > 1 and actively contributed to the aroma definition of the different wines. Among the esters, ethyl decanoate was the most abundant in the wines inoculated with *M. pulcherrima* MP346, regardless of the presence/absence of GIY. However, the differences in the VOC profiles enabled the wines produced with the different winemaking protocols to be distinguished.

The modulation of the aromatic profile of each wine was also confirmed by a sensory analysis, which made it possible to differentiate the wines into three groups. The presence of *M. pulcherrima* MP346 and the absence of GIY did not allow the T1 and T2 wines to be discriminated from a sensory profile, while these differences were greater in the C1 and C2 wines, where the only variable was represented by the addition of GIY.

Further studies are needed to evaluate the antioxidant effects of the specific inactive yeast with a guaranteed glutathione content at different times during the pre-fermentation stage (on the crushed-stemmed and drained must during the pressing stage) of Catarratto grapes.

The use of *S. cerevisiae* of a non-oenological origin, *M. pulcherrima* in the pre-fermentation stage and specific inactivated yeast with a high antioxidant power resulted in a better preservation of aromatic the compounds and colour, increasing the positive impact on the oxidative stability of the wines.

## Figures and Tables

**Figure 1 antioxidants-12-00439-f001:**
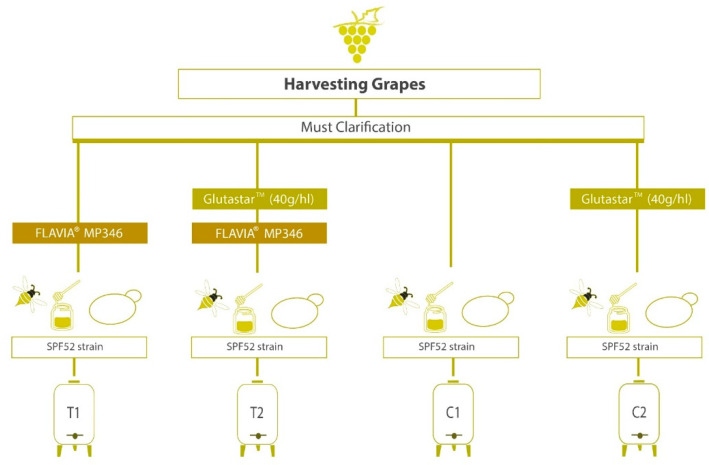
Experimental plan of wines obtained from Catarratto grape must.

**Figure 2 antioxidants-12-00439-f002:**
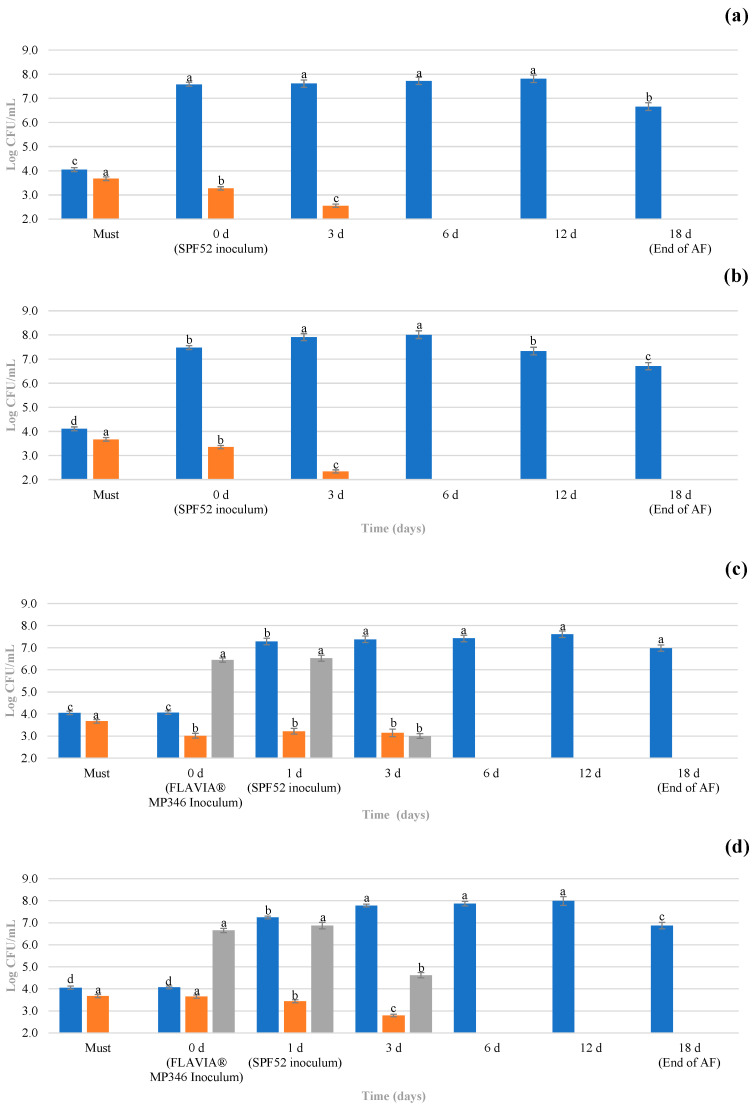
Microbiological concentration (Log CFU/mL) of samples during alcoholic fermentation: (**a**) C1, single inoculum with *S. cerevisiae* SPF52; (**b**) C2, glutathione-rich inactivated yeasts and single inoculum with *S. cerevisiae* SPF52; (**c**) T1, sequential inoculum with *M. pulcherrima* MP346/*S. cerevisiae* SPF52; (**d**) T2, glutathione-rich inactivated yeasts and sequential inoculum with *M. pulcherrima* MP346/*S. cerevisiae* SPF52. For each microbiological group, different letters indicate statistically significant values determined by using Tukey’s test (*p* ≤ 0.05). Legends: 
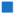
, Presumptive *Saccharomyces*; 
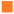
, non-*Saccharomyces* (except *Metschnikowia* spp.); 
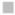
, Presumptive *Metschnikowia* spp.

**Figure 3 antioxidants-12-00439-f003:**
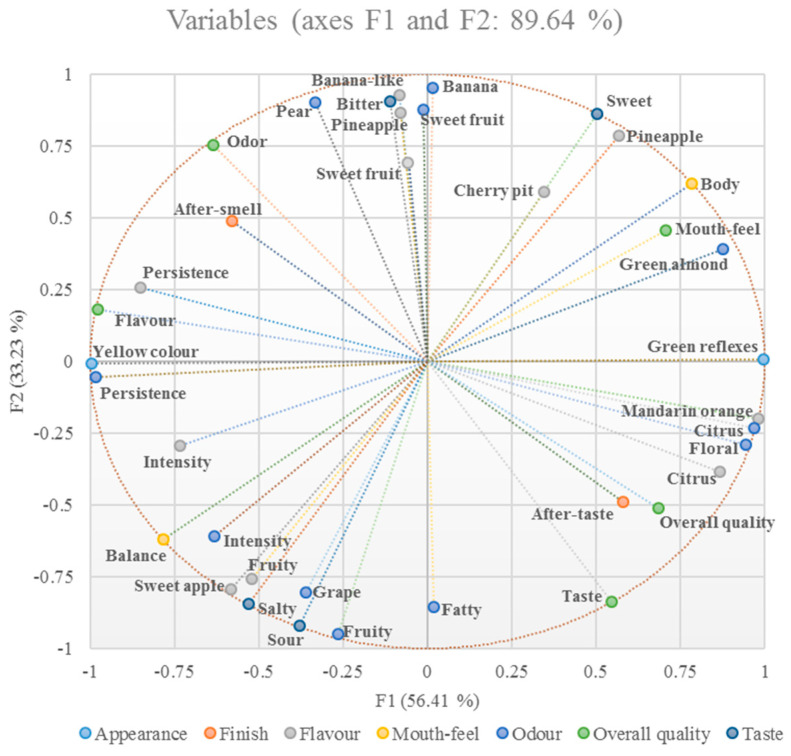
Variable loading plot of MFA.

**Figure 4 antioxidants-12-00439-f004:**
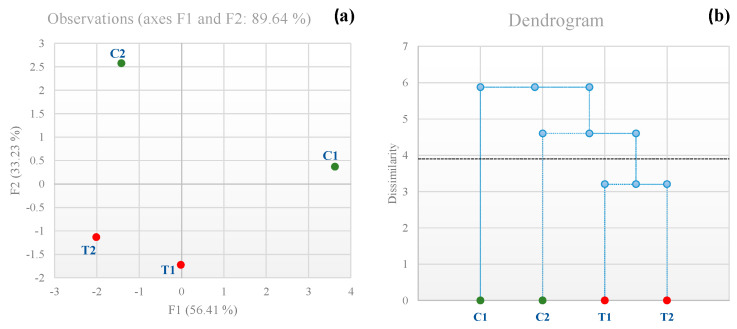
Multiple factor analysis applied to sensory analysis of Catarratto wines: (**a**) sample score; (**b**) agglomerative hierarchical clustering (AHC) dendrogram. Abbreviations: T1, sequential inoc lum with *M. pulcherrima* MP346/*S. cerevisiae* SPF52; T2, glutathione-rich inactivated yeasts and sequential inoculum with *M. pulcherrima* MP346/*S. cerevisiae* SPF52; C1, single inoculum with *S. cerevisiae* SPF52; C2, glutathione-rich inactivated yeasts and single inoculum with *S. cerevisiae* SPF52.

**Figure 5 antioxidants-12-00439-f005:**
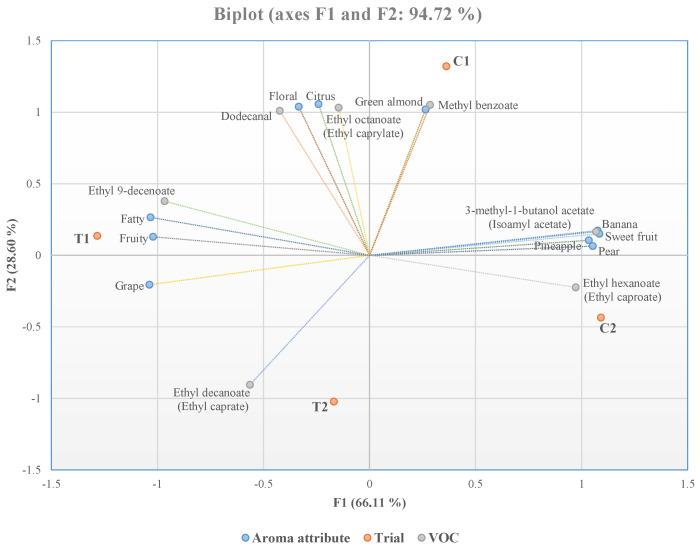
Principal component analysis (PCA) biplot for VOCs and aroma attributes. Abbreviations: T1, sequential inoculum with *M. pulcherrima* MP346/*S. cerevisiae* SPF52; T2, glutathione-rich inactivated yeasts and sequential inoculum with *M. pulcherrima* MP346/*S. cerevisiae* SPF52; C1, single inoculum with *S. cerevisiae* SPF52; C2, glutathione-rich inactivated yeasts and single inoculum with *S. cerevisiae* SPF52.

**Table 1 antioxidants-12-00439-t001:** Oenological properties of four Catarratto wines.

Sample	SO_2_ Free	SO_2_ Total	Total Extract	Total Phenols	p-DACA Flavans	Absorbance	Oxidation Test	Buffer Power	Ash Alkalinity
	(mg/L)	(mg/L)	(g/L)	(mg/L Catechins)	(mg/L Catechins)	(420 nm)	(%)	(meq/L)	(meq/L)
T1	29.00 ± 1.00 ^a^	128.00 ± 0.00 ^a^	18.80 ± 0.09 ^b^	92.74 ± 0.84 ^a^	19.80 ± 0.08 ^a^	0.079 ± 0.000 ^c^	5.74 ± 0.09 ^a^	31.25 ± 0.08 ^b^	12.28 ± 0.05 ^c^
T2	22.00 ± 0.00 ^b^	115.00 ± 2.00 ^b^	19.10 ± 0.07 ^a^	93.47 ± 0.39 ^a^	10.23 ± 0.11 ^b^	0.080 ± 0.003 ^c^	1.12 ± 0.03 ^b^	31.25 ± 0.17 ^b^	12.58 ± 0.07 ^b^
C1	16.00 ± 2.00 ^c^	109.00 ± 1.00 ^c^	18.50 ± 0.08 ^c^	84.38 ± 1.13 ^c^	1.63 ± 0.08 ^d^	0.101 ± 0.001 ^a^	0.00 ± 0.00 ^c^	31.32 ± 0.12 ^b^	12.11 ± 0.07 ^d^
C2	29.00 ± 1.00 ^a^	105.00 ± 1.00 ^d^	19.00 ± 0.11 ^ab^	83.36 ± 0.67 ^b^	3.12 ± 0.05 ^c^	0.093 ± 0.002 ^b^	0.00 ± 0.00 ^c^	32.34 ± 0.13 ^a^	13.43 ± 0.02 ^a^
S.S.	***	***	***	***	***	***	***	***	***

Result indicate mean value ± standard deviation of three determinations. Abbreviations: S.S., statistical significance; T1, sequential inoculum with *M. pulcherrima* MP346/*S. cerevisiae* SPF52; T2, glutathione-rich inactivated yeasts and sequential inoculum with *M. pulcherrima* MP346/*S. cerevisiae* SPF52; C1, single inoculum with *S. cerevisiae* SPF52; C2, glutathione-rich inactivated yeasts and single inoculum with *S. cerevisiae* SPF52. Data in the same column followed by the same letter are not significantly different according to Tukey’s test. *p* value: ***, *p* < 0.001.

**Table 2 antioxidants-12-00439-t002:** Volatile organic compounds detected in the four Catarratto wines (all values in ppb).

t_R_ (min.s)	LRI ^1^	Compounds ^2^	Aroma Description [47,48,49,50]	Odour Threshold ^3^	C1 ^4^ (OAV)	C2 ^4^ (OAV)	T1 ^4^ (OAV)	T2 ^4^ (OAV)	S.s. ^5^
		Σ Alcohols			33.00 ± 1.32 ^b^	36.48 ± 1.46 ^a^	18.84 ± 0.75 ^c^	19.19 ± 0.76 ^c^	***
10.55	758	3-methyl-1-butanol	Fusel	40,000 [51]	26.65 ± 1.07 ^b^ (<1)	29.59 ± 1.18 ^a^ (<1)	12.82 ± 0.51 ^d^ (<1)	15.59 ± 0.62 ^c^ (<1)	***
36.49	1110	Phenylethyl alcohol	Floral, rose	125,000 [52]	6.35 ± 0.25 ^b^ (<1)	6.89 ± 0.28 ^a^ (<1)	6.02 ± 0.24 ^b^ (<1)	3.60 ± 0.14 ^c^ (<1)	***
		Σ Ethers			4.75 ± 0.19 ^a^	4.16 ± 0.17 ^b^	4.24 ± 0.17 ^b^	2.53 ± 0.10 ^c^	***
32.14	1042	Ethyl benzyl ether	Tropical fruit, pineapple	unknown	4.75 ± 0.19 ^a^ (n.d. ^6^)	4.16 ± 0.17 ^b^ (n.d. ^6^)	4.24 ± 0.17 ^b^ (n.d. ^6^)	2.53 ± 0.10 ^c^ (n.d. ^6^)	***
		Σ Aldehydes			17.37 ± 0.69 ^a^	4.91 ± 0.20 ^c^	11.85 ± 0.47 ^b^	2.73 ± 0.11 ^d^	***
24.89	958	Benzaldehyde	Bitter almond, cherry	1500 [53]	6.31 ± 0.25 ^a^ (<1)	4.91 ± 0.20 ^b^ (<1)	3.60 ± 0.14 ^c^ (<1)	2.73 ± 0.11 ^d^ (<1)	***
37.08	1203	Decanal	Floral, orange peel citrus	0.1 [54]	tr (n.d. ^6^)	tr (n.d. ^6^)	tr (n.d. ^6^)	tr (n.d. ^6^)	n.d. ^6^
56.38	1411	Dodecanal	Citrus, floral	2 [55]	11.06 ± 0.44 ^a^ (5.53)	0.00 ± 0.00 ^c^ (<1)	8.25 ± 0.33 ^b^ (4.13)	0.00 ± 0.00 ^c^ (<1)	***
		Σ EEFAs			1401.74 ± 56.08 ^d^	1848.45 ± 73.94 ^c^	2318.98 ± 92.76 ^a^	2056.15 ± 82.25 ^b^	***
27.64	989	Ethyl hexanoate	Sweet fruity, pineapple, green apple	5 [55]	33.79 ± 1.35 ^b^ (6.76)	48.86 ± 1.95 ^a^ (9.77)	27.85 ± 1.11 ^c^ (5.57)	32.14 ± 1.29 ^b^ (6.42)	***
37.44	1208	Ethyl octanoate	Fruity, pear	2 [55]	901.19 ± 36.05 ^a^ (450.60)	730.52 ± 29.22 ^b^ (365.26)	837.67 ± 33.51 ^a^ (418.84)	596.78 ± 23.87 ^c^ (298.39)	***
51.00	1379	Ethyl decanoate	Fruity, grape	200 [55]	273.88 ± 10.96 ^c^ (1.37)	928.14 ± 37.13 ^b^ (4.64)	1253.71 ± 50.15 ^a^ (6.27)	1236.22 ± 49.45 ^a^ (6.18)	***
54.98	1391	Ethyl 9-decenoate	Fruity, fatty	100 [56]	184.44 ± 7.38 ^ab^ (1.84)	137.73 ± 5.51 ^c^ (1.38)	199.75 ± 7.99 ^a^ (2.00)	178.82 ± 7.15 ^b^ (1.79)	***
67.44	1599	Ethyl dodecanoate	Sweet, waxy, floral	2000 [55]	8.44 ± 0.34 ^b^ (<1)	3.20 ± 0.13 ^c^ (<1)	0.00 ± 0.00 ^d^ (<1)	12.19 ± 0.49 ^a^ (<1)	***
		Σ HAAs			15.10 ± 0.60 ^b^	19.09 ± 0.76 ^a^	6.25 ± 0.25 ^d^	9.71 ± 0.39 ^c^	***
18.59	882	3-methyl-1-butanol acetate	Sweet fruity, banana	0.75 [52]	15.10 ± 0.60 ^b^ (20.13)	19.09 ± 0.76 ^a^ (25.45)	6.25 ± 0.25 ^d^ (8.33)	9.71 ± 0.39 ^c^ (12.95)	***
		Σ EEBAs			12.94 ± 0.52 ^b^	8.02 ± 0.32 ^c^	0.00 ± 0.00 ^d^	14.56 ± 0.58 ^a^	***
58.69	1447	Isopentyl octanoate	Fruity, pineapple, coconut	125 [57]	12.94 ± 0.52 ^b^ (<1)	8.02 ± 0.32 ^c^ (<1)	0.00 ± 0.00 ^d^ (<1)	14.56 ± 0.58 ^a^ (<1)	***
		Σ *MEs*			233.83 ± 9.35 ^a^	106.27 ± 4.26 ^bc^	118.12 ± 4.74 ^b^	98.84 ± 3.96 ^c^	***
6.80	611	Ethyl acetate	Ethereal, fruity	7500 [55]	65.36 ± 2.61 ^a^ (<1)	9.10 ± 0.36 ^d^ (<1)	33.72 ± 1.35 ^c^ (<1)	38.29 ± 1.53 ^b^ (<1)	***
34.79	1089	Methyl benzoate	Green almond	10 [56]	36.94 ± 1.48 ^a^ (3.69)	25.00 ± 1.00 ^b^ (2.50)	24.22 ± 0.97 ^b^ (2.42)	14.93 ± 0.60 ^c^ (1.49)	***
46.19	1268	2-phenylethyl hexanoate	Sweet, honey, floral	94 [58]	10.28 ± 0.41 ^a^ (<1)	5.03 ± 0.20 ^b^ (<1)	0.00 ± 0.00 ^c^ (<1)	0.00 ± 0.00 ^c^ (<1)	***
46.24	1542	2-phenylethyl acetate	Rose	250 [55]	0.00 ± 0.00 ^c^ (<1)	0.00 ± 0.00 ^c^ (<1)	3.69 ± 0.15 ^b^ (<1)	5.45 ± 0.22 ^a^ (<1)	***
		Σ Others			121.25 ± 4.85 ^a^	67.62 ± 2.70 ^b^	56.49 ± 2.27 ^c^	40.17 ± 1.61 ^d^	***
7.50	634	Tetrahydrofuran	Butter, caramel	unknown	40.89 ± 1.64 ^a^ (n.d. ^6^)	35.68 ± 1.43 ^b^ (n.d. ^6^)	26.44 ± 1.06 ^c^ (n.d. ^6^)	23.34 ± 0.93 ^c^ (n.d. ^6^)	***
18.14	876	1,3-dimethylbenzene	Plastic odour	unknown	12.08 ± 0.48 ^a^ (n.d. ^6^)	8.03 ± 0.32 ^b^ (n.d. ^6^)	4.14 ± 0.17 ^c^ (n.d. ^6^)	2.89 ± 0.12 ^d^ (n.d. ^6^)	***
29.59	1023	*o*-cymene	Herb	unknown	15.37 ± 0.61 ^a^ (n.d. ^6^)	9.97 ± 0.40 ^b^ (n.d. ^6^)	5.41 ± 0.22 ^c^ (n.d. ^6^)	3.67 ± 0.15 ^d^ (n.d. ^6^)	***
34.04	1097	1-butenyl benzene	unknown	unknown	2.81 ± 0.11 ^a^ (n.d. ^6^)	2.05 ± 0.08 ^b^ (n.d. ^6^)	1.40 ± 0.06 ^c^ (n.d. ^6^)	0.76 ± 0.03 ^d^ (n.d. ^6^)	***
44.34	1232	Benzothiazole	Sulfury, rubbery, vegetable	unknown	16.45 ± 0.66 ^a^ (n.d. ^6^)	0.00 ± 0.00 ^b^ (n.d. ^6^)	0.00 ± 0.00 ^b^ (n.d. ^6^)	0.00 ± 0.00 ^b^ (n.d. ^6^)	***
50.79	1302	6-ethyltetralin (isomer)	unknown	unknown	6.85 ± 0.27 (n.d. ^6^)	3.10 ± 0.12 (n.d. ^6^)	3.44 ± 0.14 (n.d. ^6^)	tr (n.d. ^6^)	n.d. ^6^
51.29	1311	6-ethyltetralin (isomer)	unknown	unknown	7.66 ± 0.31 (n.d. ^6^)	0.00 ± 0.00 (n.d. ^6^)	2.97 ± 0.12 (n.d. ^6^)	tr (n.d. ^6^)	n.d. ^6^
54.53	1368	2-ethenyl-naphtalene	unknown	unknown	11.50 ± 0.46 ^a^ (n.d. ^6^)	6.36 ± 0.25 ^c^ (n.d. ^6^)	10.83 ± 0.43 ^a^ (n.d. ^6^)	9.51 ± 0.38 ^b^ (n.d. ^6^)	***
59.64	1485	2,6-di-tert-butylquinone	unknown	unknown	7.64 ± 0.31 (n.d. ^6^)	2.43 ± 0.10 (n.d. ^6^)	1.86 ± 0.07 (n.d. ^6^)	tr (n.d. ^6^)	n.d. ^6^

^1^ Linear retention index obtained through the modulated chromatogram reported for DB-5 MS apolar column; ^2^ compounds are classified in order of retention time; ^3^ odor threshold reported in the literature; ^4^ Relative amounts expressed as ppb with respect to calibration curve of ethyl benzoate; ^5^ statistical significance; ^6^ not determinable. Abbreviations: EEFAs: ethyl esters of fatty acids; HAAs: higher alcohol acetates; EEBAs: ethyl esters of branched acids; MEs: miscellaneous esters; OAV, odour activity value; tr: trace amount < 0.05%; T1, sequential inoculum with *M. pulcherrima* MP346/*S. cerevisiae* SPF52; T2, glutathione-rich inactivated yeasts and sequential inoculum with *M. pulcherrima* MP346/*S. cerevisiae* SPF52; C1, single inoculum with *S. cerevisiae* SPF52; C2, glutathione-rich inactivated yeasts and single inoculum with *S. cerevisiae* SPF52. Data in the same line followed by the same letter are not significantly different according to Tukey’s test. *p* value: ***, *p* < 0.001.

**Table 3 antioxidants-12-00439-t003:** Sensory score for experimental Catarratto wines.

Attributes	Trial				SEM	Statistical Significance	
	C1	C2	T1	T2		Judges	Wine
Appearance							
Yellow colour	7.28 ^a^	7.15 ^a^	7.21 ^a^	7.29 ^a^	0.01	n.s.	n.s.
Green reflexes	4.04 ^a^	3.63 ^b^	3.74 ^b^	3.68 ^b^	0.02	***	***
Odour							
Banana	3.63 ^b^	3.94 ^a^	2.79 ^d^	3.15 ^c^	0.07	***	***
Citrus	2.40 ^a^	1.00 ^c^	1.74 ^b^	1.00 ^c^	0.09	***	***
Fatty	1.35 ^b^	1.22 ^c^	1.62 ^a^	1.32 ^b^	0.02	***	***
Floral	2.53 ^a^	1.00 ^c^	1.97 ^b^	1.00 ^c^	0.10	***	***
Fruity	8.54 ^c^	8.02 ^d^	8.88 ^a^	8.68 ^b^	0.05	***	***
Grape	2.97 ^c^	2.99 ^c^	4.17 ^a^	3.43 ^b^	0.07	***	***
Green almond	7.67 ^a^	6.84 ^b^	6.77 ^b^	5.71 ^c^	0.11	***	***
Intensity	6.68 ^c^	7.19 ^b^	8.26 ^a^	7.40 ^b^	0.09	***	***
Pear	5.14 ^b^	5.44 ^a^	4.76 ^d^	4.91 ^c^	0.04	***	***
Persistence	7.11 ^d^	8.64 ^b^	8.12 ^c^	8.97 ^a^	0.10	***	***
Pineapple	3.62 ^a^	3.63 ^a^	2.96 ^c^	3.44 ^b^	0.04	***	***
Sweet fruit	7.25 ^b^	7.57 ^a^	5.75 ^d^	6.59 ^c^	0.10	***	***
Taste							
Sweet	3.48 ^a^	3.59 ^a^	2.78 ^b^	2.68 ^b^	0.06	***	***
Sour	5.38 ^b^	5.37 ^b^	8.11 ^a^	8.24 ^a^	0.21	***	***
Salty	5.70 ^c^	5.85 ^c^	7.99 ^b^	8.39 ^a^	0.18	***	***
Bitter	1.10 ^c^	1.25 ^b^	1.20 ^b^	1.42 ^a^	0.02	***	***
Mouthfeel							
Body	7.80 ^c^	8.42 ^b^	8.55 ^b^	8.97 ^a^	0.06	***	***
Balance	6.50 ^d^	7.49 ^c^	8.10 ^b^	8.65 ^a^	0.12	***	***
Flavour							
Banana-like	2.47 ^b^	2.75 ^a^	1.93 ^d^	2.22 ^c^	0.07	***	***
Cherry pit	3.67 ^a^	3.84 ^a^	3.77 ^a^	2.70 ^b^	0.07	***	***
Citrus	3.92 ^a^	1.00 ^b^	3.58 ^a^	1.00 ^b^	0.21	***	***
Fruity	6.15 ^c^	6.26 ^c^	7.79 ^a^	6.80 ^b^	0.10	***	***
Intensity	7.80 ^c^	7.85 ^c^	8.12 ^b^	8.56 ^a^	0.04	***	***
Mandarin orange	1.74 ^a^	1.00 ^c^	1.40 ^b^	1.00 ^c^	0.05	***	***
Persistence	7.70 ^c^	8.78 ^a^	7.97 ^b^	8.94 ^a^	0.08	***	***
Pineapple	7.11 ^a^	6.89 ^b^	6.86 ^b^	6.14 ^c^	0.05	***	***
Sweet apple	2.51 ^c^	2.66 ^c^	3.89 ^a^	3.54 ^b^	0.09	***	***
Sweet fruit	7.12 ^b^	7.56 ^a^	5.75 ^d^	6.58 ^c^	0.10	***	***
Overall quality	7.50 ^d^	8.57 ^b^	8.25 ^c^	8.80 ^a^	0.07	***	***
Flavour	6.98 ^c^	8.81 ^a^	8.11 ^b^	8.91 ^a^	0.11	***	***
Mouthfeel	7.20 ^c^	8.32 ^a^	7.88 ^b^	7.97 ^b^	0.06	***	***
Odour	7.20 ^c^	8.86 ^a^	8.01 ^b^	8.74 ^a^	0.10	***	***
Taste	7.01 ^d^	7.54 ^c^	7.82 ^b^	8.11 ^a^	0.06	***	***
Finish							
After-smell	6.80 ^c^	8.15 ^b^	8.21 ^b^	8.50 ^a^	0.10	***	***
After-taste	7.10 ^c^	7.96 ^b^	8.22 ^b^	8.71 ^a^	0.09	***	***

Results indicate mean value of three replicate sessions. Abbreviation: SEM, standard error of the mean; T1, sequential inoculum with *M. pulcherrima* MP346/*S. cerevisiae* SPF52; T2, glutathione-rich inactivated yeasts and sequential inoculum with *M. pulcherrima* MP346/*S. cerevisiae* SPF52; C1, single inoculum with *S. cerevisiae* SPF52; C2, glutathione-rich inactivated yeasts and single inoculum with *S. cerevisiae* SPF52. Data in the same line followed by the same letter are not significantly different according to Tukey’s test. *p* value: ***, *p* < 0.001; n.s., not significant.

## Data Availability

All data included in this study are available upon request by contacting the corresponding author.

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
