# Peer review of "Improving the Aromatic Profiles of Catarratto Wines: Impact of Metschnikowia pulcherrima and Glutathione-Rich Inactivated Yeasts"

_antioxidants, 2023, doi:10.3390/antiox12020439_

Round 1

Reviewer 1 Report (Previous Reviewer 1)

The work has been improved.

More comments

Line 141. Please provide details of the pectolytic enzymes

Line 142. Please revise no meaning.

Table S1. The column yeast inoculation does not refer to the same day of analysis. Day 0 for C and Day 1 for T. Please clarify. In the present format is not correct. And I am wondering why all of them present the same ethanol content or other parameters.

Please explain how the end of alcoholic fermentation was monitored. In addition explain why in all wines the end of fermentation was 18 days? Since different fermentation rates were observed.

Lines 358-359. Please explain this reduction in sugars during ageing. And more specifically explain why this observed after 1 month.

Line 372. “legal values”?? Please provide the authority as reference about these legal values.

Table 2. Why retention time and not Linear retention indices? According to lines 222-224 the Linear retention indices were used. The qualitative analysis was only based on MS data? Is this appropriate? Any correlation with other studies?

Figure 3. Only discussion in lines 403-406. Please delete figure. Discussion of 3 lines does not support the presence of this figure.

Sensory analysis. Please provide details. Reference 1 describes 50 attributes and the present only 36. This causes misunderstandings. In addition the scale was 0-10?

Table 3. The statistical analysis in my opinion seems misleading. For example see bitter taste. In a scale from 0-10 or 0-9 the values 1.10, 1.25, 1.20 and 1.42 results in ***?

Author Response

Answers to Reviewer 1:

R.1- The work has been improved.

AU. Thanks a lot for your general comments. Your suggestion was considered and the changes in the text were highlighted in yellow.

More comments:

R.1- Line 141. Please provide details of the pectolytic enzymes.

AU. Details of the pectolytic enzyme used have been reported in the text (L. 140-141).

R.1- Line 142. Please revise no meaning.

AU. The sentence has been changed (L. 142-143).

R.1- Table S1. The column yeast inoculation does not refer to the same day of analysis. Day 0 for C and Day 1 for T. Please clarify. In the present format is not correct. And I am wondering why all of them present the same ethanol content or other parameters.

AU. The column 'Yeast inoculum' contains the average value of all parameters measured at the time of inoculation of the starter strain S. cerevisiae SPF52. For reasons of space, we have included all values in this single column. We have decided to delete the 'Yeast inoculum' column for space reasons, leaving the 'Must' column, which gives the values of the must composition after clarification.

R.1- Please explain how the end of alcoholic fermentation was monitored. In addition explain why in all wines the end of fermentation was 18 days? Since different fermentation rates were observed.

AU. According to Bağder Elmacı et al. (2015), alcoholic fermentation is considered complete when sugar concentration in wine is less than 0.4% (4 g/L, glucose + fructose). This information is supported by legislation on dry wines (OIV, 18/73 and Eco 3/2003). In addition, the 18 day alcoholic fermentation time was considered for all trials because the residual sugar concentration did not differ significantly over two days.

R.1- Lines 358-359. Please explain this reduction in sugars during ageing. And more specifically explain why this observed after 1 month.

AU. After a month, the wine was racked to separate the coarse lees. This operation will have caused an oxygenation of the wine, allowing the reactivation of the residual yeasts and consequently the reduction of sugars.

R.1- Line 372. “legal values”?? Please provide the authority as reference about these legal values.

AU. This value can be found in the Decree of 10 August 2017 "Limits of certain components contained in wines" in application of Article 25 of Law No. 238 of 12 December 2016, published in the OG General Series No. 201 of 29 August 2017. The legislative reference has been included in the text (L. 381) and in the bibliography section (L. 703-704).

R.1- Table 2. Why retention time and not Linear retention indices? According to lines 222-224 the Linear retention indices were used. The qualitative analysis was only based on MS data? Is this appropriate? Any correlation with other studies?

AU. Dear reviewer, It was our oversight not to add the linear retention index of single compounds. Linear retention indices have now been entered in Table 2. The compounds were identified by experimental LRIs, by comparison of LRI with data present in libraries such as Adams, NIST 08, Wiley 9 and FFNSC 2, by comparison with mass spectra and by comparison with the retention times of pure compounds such as benzaldehyde, decanal, dodecanal, phenylethyl alcohol, ethyl caprylate, ethyl caprate, and methyl benzoate.

R.1- Figure 3. Only discussion in lines 403-406. Please delete figure. Discussion of 3 lines does not support the presence of this figure.

AU. Figure 3 and the related discussion in the text have been removed. As a result, the numerical order of the figures has changed (L. 509; L. 514; L: 516-517; L. 521; L. 528; L. 547).

 R.1- Sensory analysis. Please provide details. Reference 1 describes 50 attributes and the present only 36. This causes misunderstandings. In addition the scale was 0-10?

AU. Sorry for the misunderstanding. The difference of 14 descriptors is due to the different biotechnology used while maintaining the same matrix (Catarratto grapes) compared to reference 1. In any case, a sentence was included in the text describing at an early stage how the judges selected the 36 attributes (L. 233-236). Furthermore, the scale used was 1-9 (L. 244-245) and not 0-10. The scale of 0-10 in reference 1 was used to describe overall acceptability in the Acceptance test.

R.1- Table 3. The statistical analysis in my opinion seems misleading. For example see bitter taste. In a scale from 0-10 or 0-9 the values 1.10, 1.25, 1.20 and 1.42 results in ***?

AU. The standard deviations of the values defining the bitter attribute in each trial are low. This is also justified by the low SEM value in Table 2. We repeated the statistical analysis for all values reported in the table and found no errors.

Reviewer 2 Report (New Reviewer)

It is an interesting study for improving the aromatic profiles of Catarratto wines. However it would be better mayor revision to develop the target of the study non-Saccharomyces/Saccharomyces sequential inoculation to achieve a positive impact in the aroma. Please re-write the manuscript using paragraphs to separate each important point.

The abstract should be re-write in order to summarize the work and specially focus on the purpose of the study (hypothesis, overall question, objective). Please remove the commercial name (ie. GlutastarTm,  FlaviaR MP346, etc) that should be described at the beginning of material and methods before the experimental design. It is very useful in a commercial brochure but no in a scientific paper. It would be easier to understand when the discussion was around non-Saccharomyces/Saccharomyces and glutathione-rich inactivated yeasts.

Specify points:

-       “Saccharomyces species” should be better “Saccharomyces spp”

-       2.5. “Dominance of S. cerevisiae and M. pulcherrima isolates” should be in italics letter

-       Chemical compounds should be write in lowercase letter, such for example “2-Phenylethyl…

-       Ln 275 Metschnikowia pulcherrima should be M. pulcherrima

-       Legends of figure and tables should be contained all the information, please included identification of treatements and samples (C1, C2, T1, SS…)

-       It is more common used the term “Microbial load” instead of “microbial concentration”

-       -Please include AHC description in statistical analysis

-       -“Chemical” or “Physico-chemical properties” instead of “parameters”. Parameters should be pressure, temperature, etc.

-       -N.S in lowercase like in legend (n.s.)

-       -“data in the same row or column” instead of “data within a line”

Author Response

Answers to Reviewer 2:

R.2- It is an interesting study for improving the aromatic profiles of Catarratto wines. However it would be better mayor revision to develop the target of the study non-Saccharomyces/Saccharomyces sequential inoculation to achieve a positive impact in the aroma. Please re-write the manuscript using paragraphs to separate each important point.

AU. Your suggestion was considered and the changes in the text were highlighted in green. The requested information was added to the introduction following the suggestions required by reviewer 1.

R.2- The abstract should be re-write in order to summarize the work and specially focus on the purpose of the study (hypothesis, overall question, objective). Please remove the commercial name (ie. GlutastarTm,  FlaviaR MP346, etc) that should be described at the beginning of material and methods before the experimental design. It is very useful in a commercial brochure but no in a scientific paper. It would be easier to understand when the discussion was around non-Saccharomyces/Saccharomyces and glutathione-rich inactivated yeasts.

AU. The abstract was rewritten in relation to the suggestions proposed (L. 18-32). Throughout the text, 'FLAVIA® MP346' has been replaced by 'M. pulcherrima MP436'. While 'GlutastarTM' has been changed to 'GIY', an acronym for glutathione-rich inactivated yeasts.

R.2- “Saccharomyces species” should be better “Saccharomyces spp”

AU. Done (L. 46).

R.2- 2.5. “Dominance of S. cerevisiae and M. pulcherrima isolates” should be in italics letter

AU. Done (L. 178).

R.2- Chemical compounds should be write in lowercase letter, such for example “2-Phenylethyl…

AU. All VOC names have been corrected in the text (L. 194; L. 426; L. 439-440; L. 449). Suggestions have also been made in Table 2 and Figure 5.

R.2- Ln 275 Metschnikowia pulcherrima should be M. pulcherrima

AU. Done (L. 271).

R.2- Legends of figure and tables should be contained all the information, please included identification of treatements and samples (C1, C2, T1, SS…)

AU. Done (L. 294-297; L. 372-374; L. 444-446; L. 522-524; L. 549-551).

R.2-   It is more common used the term “Microbial load” instead of “microbial concentration”

AU. Done (L. 279).

R.2- Please include AHC description in statistical analysis

AU.In section 2.8 this information has been included (L. 255-257).

R.2- “Chemical” or “Physico-chemical properties” instead of “parameters”. Parameters should be pressure, temperature, etc.

AU. Done (L. 183-184; L. 249; L. 327; L. 364-365; L.368; L.370; L.581; Table S1).

R.2- N.S in lowercase like in legend (n.s.)

AU. Done.

R.2- data in the same row or column” instead of “data within a line”

AU. Done.

Round 2

Reviewer 2 Report (New Reviewer)

Table 1: divided in two tables one with "alcoholic fermentation time (days)": 3, 6, 12, 18, another table with "Steel aging time (month)": 1,3, 5.

Bottling was done after 5 months of stell aging? If there is not more additional storage time, no new datas are obtained in the step of bottling. Please remove this datas or give more information time and discussion in the manuscript.

Author Response

R.2- Table 1: divided in two tables one with "alcoholic fermentation time (days)": 3, 6, 12, 18, another table with "Steel aging time (month)": 1,3, 5.

AU. I think the reviewer is referring to Table S1. In any case, the suggestions were applied. This resulted in there being two tables in the supplementary material: S1 and S2. The application of the suggestions led to changes in the text of the manuscript (L.364; L. 581-583).

R.2- Bottling was done after 5 months of stell aging? If there is not more additional storage time, no new datas are obtained in the step of bottling. Please remove this datas or give more information time and discussion in the manuscript.

AU. Bottling was carried out at the 5th month. Chemical parameters at bottling were removed.

This manuscript is a resubmission of an earlier submission. The following is a list of the peer review reports and author responses from that submission.

Round 1

Reviewer 1 Report

1.The work is well prepared with defined objective. However, the use of M. pulcherrima and S. cerevisiae sequential inoculation for wine-making is very popular and the authors should clearly presents the novelty of their work also at the end of introduction. Some specific comments:

2.       Lines 40-57. Some recent works/review are missing. Please also use the following to increase the impact of discussion Lappa, I. K., Kachrimanidou, V., Pateraki, C., Koulougliotis, D., Eriotou, E., & Kopsahelis, N. (2020). Indigenous yeasts: emerging trends and challenges in winemaking. Current Opinion in Food Science, 32, 133-143. https://doi.org/10.1016/j.cofs.2020.04.004 and Gonzalez, R., & Morales, P. (2022). Truth in wine yeast. Microbial Biotechnology, 15(5), 1339-1356. https://doi.org/10.1111/1751-7915.13848

3.       Lines 58-67. Please add here and discuss similar recent works with M. pulcherrima and S. cerevisiae sequential inoculation for wine-making.

4.       Line 71. Please also use a review with the role of glutathione in wine-making . Kritzinger, E. C., Bauer, F. F., & Du Toit, W. J. (2013). Role of glutathione in winemaking: a review. Journal of agricultural and food chemistry, 61(2), 269-277. https://doi.org/10.1021/jf303665z . Also add more discussion here by adding some recent studies with the application/evaluation of glutathione in wine-making.

5.       Line 149 Please replace FA to AF.

6.       Section 2.6.1. All analyses refer to ref 1. However, since in ref 1 there is a link to other study for the analyses this is inappropriate, in my opinion. Since the analyses were not made according to ref 1 but according to another reference reported in ref 1 the original ref should be reported (e.g. OIV, other manuscripts etc.).

7.       Section 2.7. Please add more information about the judges (age, gender etc.)

8.       Figure 2. It is not obvious what the different lines (colors) represent. Please clarify also in the figure.

9.       Lines 243-256. It would be better to have also the results of chemical analyses in order to support these comments. For example ethanol content, acidity etc.

10.   Figure 4a. It would be better to presented as single figure. In the present format it is difficult to read.

11.   Line 513. Delete were (two times)

12.   Conclusions. More conclusions are needed and especially at the end some future studies required in the field and also the significance of the results of the present study.

Reviewer 2 Report

The paper "Improving the aromatic profiles of Catarratto wines: impact of Metschnikowia pulcherrima and glutathione-rich inactivated yeasts" reports the chemical and sensory results from vinifications of Catarratto white grape carried out under different microbiological and wine chemical conditions. Some results are of some interest to the field but some should be better explained or revised. Some experimental conditions are not clearly exposed. Many comparisons between the chemical values in the paper and others reported in other previuous papers are speculative as no explanation of the meaning is reported.

Many major points must be revised.
